# Nicotine Neurotoxicity Involves Low Wnt1 Signaling in Spinal Locomotor Networks of the Postnatal Rodent Spinal Cord

**DOI:** 10.3390/ijms22179572

**Published:** 2021-09-03

**Authors:** Jaspreet Kaur, Graciela L. Mazzone, Jorge B. Aquino, Andrea Nistri

**Affiliations:** 1Department of Neuroscience, University of Copenhagen, 2200 Copenhagen N, Denmark; 2Department of Neuroscience, International School for Advanced Studies (SISSA), 34136 Trieste, Italy; nistri@sissa.it; 3Instituto de Investigaciones en Medicina Traslacional (IIMT), CONICET-Universidad Austral, Av. Pte. Perón 1500, Pilar B1629AHJ, Buenos Aires, Argentina; jaquino@austral.edu.ar

**Keywords:** nicotine toxicity, Wnt1 pathway, spinal cord injury, locomotor networks, excitotoxicity, fictive locomotion, postnatal, rat, mice

## Abstract

The postnatal rodent spinal cord in-vitro is a useful model to investigate early pathophysiological changes after injury. While low dose nicotine (1 µM) induces neuroprotection, how higher doses affect spinal networks is unknown. Using spinal preparations of postnatal wild-type Wistar rat and Wnt1Cre2:Rosa26Tom double-transgenic mouse, we studied the effect of nicotine (0.5–10 µM) on locomotor networks in-vitro. Nicotine 10 µM induced motoneuron depolarization, suppressed monosynaptic reflexes, and decreased fictive locomotion in rat spinal cord. Delayed fall in neuronal numbers (including motoneurons) of central and ventral regions emerged without loss of dorsal neurons. Conversely, nicotine (0.5–1 µM) preserved neurons throughout the spinal cord and strongly activated the Wnt1 signaling pathway. High-dose nicotine enhanced expression of S100 and GFAP in astrocytes indicating a stress response. Excitotoxicity induced by kainate was contrasted by nicotine (10 µM) in the dorsal area and persisted in central and ventral regions with no change in basal Wnt signaling. When combining nicotine with kainate, the activation of Wnt1 was reduced compared to kainate/sham. The present results suggest that high dose nicotine was neurotoxic to central and ventral spinal neurons as the neuroprotective role of Wnt signaling became attenuated. This also corroborates the risk of cigarette smoking for the foetus/newborn since tobacco contains nicotine.

## 1. Introduction

Transient application of nicotine (Nic) to brain and spinal motor networks in vitro can robustly protect them from excitotoxicity and neurodegeneration [1,2,3]. In particular, using the isolated rodent spinal cord maintained in vitro for 24 h as a model of acute spinal injury [4,5], we have observed that nicotine evokes a neuroprotective action against excitotoxic damage induced by the glutamate agonist kainate; KA [2]. The translational implication of these data suggests that the role of nicotine might extend beyond its well-known effect of alleviating certain forms of chronic pain in humans [6], a phenomenon at least in part attributable to positive modulation of inhibitory synaptic transmission in the spinal cord [7,8,9]. Nevertheless, acute administration of nicotine to smokers with spinal cord injury (SCI) induces a sharp increase in neuropathic pain [6], demonstrating a dual action of nicotine. Thus, in a spinal cord model, we set out to investigate the borderline between neuroprotection and toxicity exerted by nicotine to discover mechanisms that could contrast this transition. In particular, although the neurotoxicity of nicotine is widely documented [10], especially in young people [11], little is known about its possible toxic effects on spinal networks. Although high-dose nicotine is a well-known convulsive agent that activates brain centers [12,13], whether nicotine can cause hyperexcitability or depression of spinal circuitries is unclear.

In the mouse hippocampus, during the process of neuroprotection against various insults, nicotine activates Wnt (wingless-related MMTV integration site) signaling pathways which in turn upregulate α7 nicotinic acetylcholine receptors (nAChRs) via transcriptional activation of Wnt target genes related to survival [14]. Either short (30 min) or 24 h application of nicotine is reported to positively regulate Wnt via a PKC-dependent process in cultured cells [15].

The Wnt/β-catenin signaling pathway comprises a family of secreted lipid-modified glycoproteins involved in cell–cell communication through intercellular signaling [16,17], with an essential role in vertebrate development [18,19] including spinal cord dorsal–ventral patterning [20]. Furthermore, the direction of axon movement and specific innervation depends on Wnt signaling mechanisms [19,21]. Wnt1 and Wnt3a are expressed in extensively overlapping regions within the central nervous system, predominantly along the dorsal midline from the diencephalon to the spinal cord, and are required for the specification of dorsal interneurons [20]. The signaling pathways involved in neuronal recovery and regeneration following spinal cord and peripheral nerve injury have garnered growing attention. Within this framework, an in vitro study has shown that inactivation of Wnt signaling depresses spinal excitatory synaptic transmission [22]. An increase in the expression levels of Ryk, a Wnt receptor, was found a few hours after contusion of the rat spinal cord, consistent with early involvement of this signaling pathway in the pathophysiology of SCI [23]. Furthermore, recent studies have demonstrated that, after blocking Wnt/Ryk receptor signaling, progression of neuropathic pain due to spinal nerve ligation is reduced [24]. Wnts can also modulate astrocyte function [25] and contribute to the formation of glial scars after SCI in adult rats [26]. Wnt1 is reported to promote the expression of the glial glutamate transporter EAAT2, which has been associated with astrocyte-mediated protection in cultured dopaminergic cells through the decrease in the concentration of extracellular glutamate [27]. Therefore, it appears that the Wnt/β-catenin signaling pathway may contribute to certain neuroprotective effects as, in a rat SCI model, its induction results in significant locomotion improvements [28]. Thus, it seemed interesting to study changes in the expression of Wnts in the spinal cord tissue in response to an acute treatment with nicotine or KA (as an experimental excitotoxic stimulus) or a combination of both.

Hence, the aims of the present study were to investigate the effect of high-dose nicotine (10 µM), as well as of its interaction with KA, on synaptic transmission and rhythmic output of motor networks, neuronal and astroglia numbers, and the activation of the Wnt1 promoter. To this end, we used in vitro spinal cord preparations of wild-type rat as well as transgenic mice expressing Wnt1Cre2:Rosa26Tom (here Cre represents Cre recombinase enzyme) for interrogating gene function and activity. The Wnt1Cre2 transgenic mouse line includes the Wnt1 promoter and enhancer sequences without the gene sequence itself. Indeed, crossing this transgenic mouse with Rosa26Tom reporter mouse produces a useful model for studying the differentiation of neuronal subtypes and drive gene expression to design restorative therapies [29].

## 2. Results

### 2.1. Dose-Dependent Effect of Nicotine on Network Depolarization, Wnt1 Pathway, and Neuronal Numbers

Figure 1A shows representative examples of the ventral root (VR; left L5) depolarization (representing the integrated output of motoneuron pools) evoked by nicotine applied at various concentrations up to 10 μM (Nic 10). The bar chart of Figure 1B shows that Nic 10 induced significant depolarization that persisted for several min as exemplified in Figure 1A. Statistical analysis confirmed that the depolarization response to 10 µM nicotine was the strongest one as indicated by the fold increase in Nic 10 amplitude in comparison to baseline (Nic 10 vs. baseline: 211; Nic 2 vs. baseline: 120; Nic 1 vs. baseline: 900; Nic 0.5 vs. baseline: 10) and *p* values (Nic 10 vs. Baseline: *** *p* ≤ 0.001, *t* = 8.416; Nic 2 vs. Baseline: *p* = 0.001, *t* = 4.862; Nic 1 vs. Baseline: ** *p* = 0.004, *t* = 4.033; Nic 0.5 vs. Baseline: *p* = 0.004, *t* = 4.1, Holm–Sidak Method). Furthermore, the highest nicotine dose also elicited a flurry of high-frequency discharges indicating repeated firing of motoneurons. Although 10 µM nicotine has previously been shown to exert neuroprotective effects on hypoglossal motoneurons in rats [1], the extent and duration of such effect on spinal networks suggested its potential neurotoxicity. To further investigate this phenomenon which presumably developed slowly, the effect of different nicotine concentrations on the number of neurons (immunolabeled for the specific NeuN marker) and the activation of Wnt1 were analyzed 24 h later by using slices from the lumbar (L) spinal cord of Wnt1Cre2:Rosa26Tom mice. The application of 0.5 and 1 μM nicotine for 4 h exhibited a neuroprotective effect on such neurons in the ventral horn as their number was higher than in sham preparations 24 later, thus, suggesting that this drug had likely slowed down the standard neuronal loss usually occurring during long in vitro maintenance. Conversely, the neuronal number was significantly reduced 24 h after Nic 10 administration (Figure 2A, upper panel; sham vs. Nic 10: *p* = 0.01, U = 53.5, Mann–Whitney test; 24% decrease). Application of 0.5 μM of nicotine largely increased Wnt1 signal compared to sham (85% increase), an effect observed at a lesser extent when higher doses (34–47% increase) were applied (Figure 2A, lower panel, and B; sham vs. Nic 0.5: *p* ≤ 0.001, *t*_17.487_ = −6.83, Welch’s *t-*test; sham vs. Nic 1: *p* ≤ 0.001, *t*_24.845_ = −3.808, Welch’s *t-*test; sham vs. Nic 10: *p* ≤ 0.001, *t*_31.645_ = −8.592, Welch’s *t-*test). Appendix A shows, in a transverse section of the mouse spinal cord, broad expression of the Wnt1 signal in the blood vessels, by colocalization of CD31 antibody labeling [30]. The Pearson correlation coefficient (PCC) was used to quantify the degree of colocalization between Wnt1 signal with a neuronal (NeuN), glial (S100) or blood vessels (CD31) marker [31]. Costes et al. [32] methodology approach for automatically identifying the threshold value was used to identify background based on an analysis that determines the range of pixel values for which a positive PCC was obtained by using FiJi software. The analysis of the Wnt1 signal in fluorescence microscopy images over the entire ventral spinal region indicated high distribution in blood vessels (Appendix A) with sample images of positive signals for both probes shown in white. When red and green pixel intensities were quantified by PCC, a significant increase was observed for CD31 (0.87 ± 0.01) vs. S100 (0.41 ± 0.05) and vs. NeuN (0.20 ± 0.07), respectively (S100 vs. CD31: *p* < 0.004; NeuN vs. CD31: * *p* < 0.047; Dunn’s method).

These results are consistent with previous reports showing a neuroprotective effect of 1 μM nicotine to spinal locomotor networks [2] and suggest a narrow range of nicotine levels near the border between a neuroprotective and a toxic effect.

To support a possible involvement of the Wnt1 signaling pathway, the Wnt1 antagonist irinotecan was applied at a concentration of 5 µM [33,34] (Figure 2C) and was found to partially inhibit the expression of Wnt1-Cre-Tom compared to sham (12% decrease) and prevent the rise elicited by 0.5 (Nic 0.5 vs. Nic 0.5+Irinotecan: 37% decrease) or 10 μM (Nic 10 vs. Nic 10+Irinotecan: 53% decrease) nicotine on the activation of the Wnt1 promoter (*p* < 0.001, F_5,121_ = 14.47, one-way analysis of variance test).

### 2.2. Effect of High Nicotine Levels on Rat Spinal Neurons and Motoneurons

To investigate the topology of the effect of a high dose of nicotine on rat spinal neurons and motoneurons, neuronal (NeuN+) numbers from the dorsal (D), central (C), and ventral (V) regions and ventral motoneurons (immunostained for SMI32) were quantified. After nicotine application (4 h), no change was found in the number of dorsal neurons 24 h later; however, a significant decrease was observed in regions C and V (Figure 3A,B; *p* ≤ 0.001, F_5,53_ = 127.77, One Way Analysis of Variance test; C: 27% decrease; V: 21% decrease) and among ventral motoneurons (Figure 3A,C; 26% decrease), compared to sham. These data suggest a neurotoxic effect of high nicotine on certain spinal neurons and motoneurons, particularly in the regions believed to be essential for generating the locomotor rhythm [35,36].

### 2.3. Upregulation of S100 and GFAP by Nicotine 10 μM

A strong increase in GFAP expression in the fetal brain has previously been reported after exposure to nicotine [37]. Here, similar results were obtained in the spinal cord of newborn rodents 24 h after the application of Nic 10 for 4 h. Astroglia response to treatment with Nic 10 was monitored with S100 (Figure 4A, upper panel; sham: *n* = 10; Nic 10: *n* = 7) and GFAP (Figure 4A lower panel; sham: *n* = 6; Nic 10: *n* = 4) immunostaining in postnatal rat spinal preparations. The Nic 10 treatment induced an overexpression of S100 in all spinal regions D (34% increase), C (27% increase), and V (32% increase) (*p* ≤ 0.001, F_5,50_ = 10.087, one-way analysis of variance test). Even though no significant changes were observed in GFAP expression levels in the D region, this signal was increased in the C (46% increase) and V (60% increase) regions (*p* = 0.023, F_5,29_ = 3.217, one-way analysis of variance test). S100 expression was also analyzed in the spinal cord of Wnt1Cre2:Rosa26Tom mice, with an overall enhanced immunofluorescence intensity (Figure 4B) (*p* = 0.044, *t*_12.98_ = -2.224, Welch’s *t-*test; 21% increase). These data suggest that Nic 10 induced a stress-like response probably through astrocyte differentiation likely due to over-activation of nAChRs caused by high-dose nicotine [37,38]. 

### 2.4. Impaired Monosynaptic Transmission and Fictive Locomotion Induced by Nic 10

After treatment with Nic 10, synaptic transmission was recorded 24 h later from VRs of the rat L spinal cord and compared with that of sham preparations. Monosynaptic transmission (Figure 5A) was substantially impaired after incubation with nicotine (*p* ≤ 0.001, U = 0, Mann–Whitney test; 81% decrease); however, the polysynaptic reflex was not significantly depressed (Figure 5B; amplitude: *p* = 0.445, U = 65; area: *p* = 0.617, U = 70; Mann–Whitney test). 

Pulse trains applied to a single L dorsal root (DR) are known to evoke segmentally alternating oscillations on top of the cumulative depolarization observed on VRs. These oscillations have the hallmarks of electrically induced fictive locomotion [39] and their number was significantly reduced after treatment with Nic 10 (*p* ≤ 0.001, *t*_22.66_ = 3.943, Welch’s *t-*test) (Figure 5C), while no depression was observed for the cumulative depolarization amplitude (*p* = 0.551, *t*_19.56_ = 0.607, Welch’s *t-*test) or its area (*p* = 0.929, *t*_19.604_ = 0.0904, Welch ’s *t-*test) (Figure 5C). Fictive locomotor patterns alternating among flexor and extensor motor pools were also elicited by bath-applied drugs such as N-methyl-D-aspartate (NMDA) plus 5-hydroxytryptamine (5HT) [40]. In the present study, Nic 10 significantly decreased the cycle amplitude by 56% of the chemically induced fictive locomotion (amplitude: *p* = 0.001, *t*_6.588_ = 5.548; Welch’s *t-*test) (Figure 5D) with no changes in the cycle periodicity (period: *p* = 0.09, *t*_13_ = −1.801; Welch’s *t-*test).

### 2.5. Effect of Nicotine (10 μM) on Dorsal–Dorsal Root Potential

To further investigate whether Nic 10 was able to affect the functionality of rat dorsal horn neurons, the dorsal–dorsal root potentials (D-DRPs) were recorded from one DR following stimulation of an adjacent ipsilateral DR in sham or Nic 10 conditions (Appendix A). Despite the slightly smaller average amplitude of this response after Nic 10, there was no significant difference with the sham control, which is consistent with our previous results suggesting that there was no loss of dorsal horn neurons (*p* = 0.106, U = 83, Mann–Whitney test).

### 2.6. Neuronal Number, Motoneurons and Wnt1 Pathway after Excitotoxic Insult Followed by Nic 10

As expected, application of KA (50 μM; 1 h) induced excitotoxicity causing 24 h later substantial neuronal death (NeuN+ neurons; *p* ≤ 0.001, F_8,80_ = 159.745, one-way analysis of variance test) in the D (*p* ≤ 0.001, *t*_15.2_ = 9.337, Welch’s *t-*test; 51% decrease in neuron number), C (*p* ≤ 0.001, *t*_14.36_ = 4.327, Welch’s *t-*test; 20% decrease) and V (*p* ≤ 0.001, *t*_15.70_= 8.307, Welch’s *t-*test; 38% decrease) regions of the rat spinal cord. Interestingly, Nic 10 prevented the fall in neuronal numbers evoked by KA in the D region (KA vs. KA+Nic/Nic 4 h: *p* < 0.001, *t*_15.03_ = −10.21, Welch’s *t-*test; 91% higher); however, Nic 10 was unable to counteract excitotoxic damage in spinal regions C and V (Figure 6A, upper panel histogram). The D-DRPs, that were abolished by KA (87% decrease by KA in comparison Nic 10), persisted one day later when Nic 10 was applied together with KA followed by Nic 10 for 4 h (KA+Nic/Nic 4 h; Appendix A; *p* ≤ 0.001, H_(2)_ = 30.156, Kruskal–Wallis one-way analysis of variance on ranks test).

The same Nic 10 dose could not protect ventrally located neurons/motoneurons, when applied together with KA followed by nicotine for another 4 h (Figure 6A, lower panel histogram; KA vs. KA+Nic/Nic 4 h: *p* = 0.2, *t*_41.96_ = −1.248, Welch’s *t-*test). Similar results were obtained with Wnt1Cre2:Rosa26Tom mouse preparations by applying nicotine (10 μM) together with KA, followed by nicotine (4 h) (Figure 6B, upper panel; KA vs. KA+Nic/Nic 4 h: *p* = 0.42, *t*_26_ = −0.83, Welch’s *t-*test) as nicotine could not alter the ventral neuronal loss elicited by KA. 

The administration of KA alone did not induce changes in Wnt1 signal compared to sham (Figure 6B, lower panel; sham vs. KA: *p* = 0.910, *t*_22.052_ = 0.114, Welch’s *t-*test). This suggests that KA could not modulate the activation of the Wnt1 promoter in the ventral spinal cord. Nevertheless, the application of Nic 10 together with KA resulted in an overall reduction in Wnt1 signal (Figure 6B, lower panel; KA vs. KA+Nic/Nic 4 h: *p* = 0.005, *t*_21.2_ = 3.13, 23% decrease; sham vs. KA+Nic/Nic 4 h: *p* ≤ 0.001, *t*_21.857_ = 5.073, Welch’s *t-*test, 24% decrease). Such a decrease in the expression of the reporter gene reached a similar level to the one found when applying the Wnts pathway inhibitor irinotecan (see Figure 2C), suggesting a likely inhibition of the Wnts signaling pathway by Nic 10. To further explore any relation between nicotine toxicity, Wnts pathway, and excitotoxicity, we counted the global number of pyknotic cells as a reliable index of global cell death in our model [4]. Thus, in the ventral region, KA significantly increased the occurrence of pyknotic nuclei (observed with DAPI staining, Appendix A; sham vs. KA: *p* ≤ 0.001, *t*_15.31_= −10.629, Welch’s *t-*test, 58-fold increase), a process already detectable at the end of 1 h application of KA and only slightly less intense after KA+Nic 10 followed by Nic 10 for 4 h (KA vs. KA+Nic/Nic 4 h: *p* = 0.003, *t*_29.975_ = 3.195, Welch’s *t-*test, 41% decrease). In keeping with data shown in Figure 2, 3 Nic 10 administration evoked a significant rise in the number of pyknotic nuclei (sham vs. Nic 10: *p* = 0.03, *t*_8.15_ = −2.61, Welch’s *t-*test, 15-fold increase). After application of irinotecan (Appendix A), the number of pyknotic cells was similar to untreated controls, indicating that in basal conditions Wnts pathway activity was not constitutively operating to support cell survival in vitro, whereas nicotine administration together with irinotecan drastically increased the number of pyknotic cells (sham vs. Irinotecan+Nic 10: *p* ≤ 0.001, *t*_18.37_ = −9.58, 52 fold increase; Irinotecan vs. Irinotecan+Nic 10: *p* ≤ 0.001, *t*_18.496_ = −9.435, Welch’s *t-*test).

### 2.7. Poor Recovery in Reflex Responses and Fictive Locomotion

The monosynaptic reflex amplitude was equally depressed 24 h after Nic alone (cf. Figure 5A), KA or Nic 10 after KA followed by nicotine (4 h) (Figure 7A; *p* = 0.2, H_(2)_ = 3.108, Kruskal–Wallis one-way analysis of variance on ranks test). Conversely, polysynaptic reflexes were partly improved (in amplitude by 92% and area by 109%; KA vs. KA+Nic/Nic 4 h) after the combined treatment with Nic 10 and KA vs. KA alone (Nic 10 vs. KA: amplitude: 73% decrease; area: 85% decrease) (Figure 7B; amplitude: *p* ≤ 0.001, F_2,56_ = 32.99; area: *p* ≤ 0.001, F_2,55_ = 57.15; one-way analysis of variance test). Both electrically and chemically induced fictive locomotor patterns were completely abolished by KA. The standard protocol of Nic 10 plus KA compared with KA alone did not improve any component of the DR-induced response including cumulative depolarization and oscillations (Figure 7C; number of oscillations: *p* ≤ 0.001, F_2,33_ = 51.9; cumulative depolarization: *p* ≤ 0.001, F_2,40_ = 21.1; area: *p* ≤ 0.001, F_2,40_ = 31.26; one-way analysis of variance test). Nonetheless, in the case of chemically induced fictive locomotion, low amplitude slow oscillations were observed after such a protocol (Figure 7D; amplitude: *p* ≤ 0.001, H_(2)_ = 15.262; period: *p* ≤ 0.001, H_(2)_ = 14.563; Kruskal–Wallis one-way analysis of variance on ranks test).

## 3. Discussion

### 3.1. An Evolving Scenario from Neuroprotection to Neurotoxicity

The key finding of this study is that nicotine-induced neurotoxicity in the spinal neuronal network of newborn rodents, and that there was a fine dividing line between neuroprotective (1 µM) [2] and neurotoxic (10 µM) doses of this drug. Both effects were associated with various degrees of motor pool depolarization which lingered for several minutes. After 10 µM nicotine, intense motoneuron firing appeared and was, 24 h later, followed by neuronal loss in the central and ventral areas of the spinal cord. This phenomenon had the main characteristics of excitotoxicity which had replaced the neuroprotective action earlier reported [2]. The onset of nicotine toxicity occurred in parallel with a reduction in the ability of nicotine to activate the Wnt1 signaling pathway in the spinal cord (Figure 8). Our genetic model to induce Tomato gene expression in the spinal cord by crossing Wnt1-Cre with Rosa26-Tom mice allowed us to monitor functional changes resulting from the activation of Wnt1 mediated by different nicotine concentrations. While low doses of nicotine led to an increased number of surviving neurons in spinal slices and raised Wnt1 activity, neurotoxicity was accompanied by lower Wnt1 activity (Figure 8). These observations imply that neuronal resilience and Wnt1 stimulation proceeded in parallel, although there was no implicit suggestion of causality between these two events because it is not known whether a few surviving neurons strongly expressed Wnt1 or a larger number of neurons moderately activated Wnt1. Indeed, the correlation between localization of Wnt1 activity and neurons was weak, indicating a range of potential interpretations like the nicotine-evoked release of unidentified neuronal messengers stimulating Wnt1 in non-neuronal cells, the release of other Wnt proteins by neurons to affect Wnt1 expressing cells, or nicotine desensitization occurring preferentially at certain cholinergic receptors. These notions remain currently conjectural and will require future investigation. Nevertheless, pharmacological inhibition of Wnt signaling by irinotecan [33,34] largely increased the extent of neuronal pyknosis, a marker of cell death, thus consistent with a Wnt role in this process.

Notwithstanding elucidation of these issues, it was apparent that nicotine toxicity strongly depressed monosynaptic transmission on motoneurons while displaying much less depression of polysynaptic transmission and cumulative depolarization evoked by repeated DR stimuli. The preferential inhibition of monosynaptic reflexes suggests an effect targeted at the presynaptic level of transmitter release [41] which could be, at least partly, overcome by stronger afferent fiber stimulation and consequently stronger release typically observed with polysynaptic reflexes. It was, however, clear that perturbation by nicotine (10 µM) of premotor networks led to disruption of fictive locomotion which is a highly integrated function necessitating complex, concerted activity at the level of the central pattern generator [35,36,42]. Thus, chemically-driven fictive locomotion that requires broad recruitment of locomotor circuits by bath-applied drugs was depressed, whereas electrically-induced fictive locomotion that is based on a more segmental recruitment was suppressed. We propose that strong excitation of motor networks by Nic 10 was the initial step to produce synaptic depression and cell death.

Previous studies have demonstrated how prenatal or postnatal exposure to nicotine leads to disruption of synaptic transmission within brainstem respiratory motor networks by altering GABAergic [43], glycinergic [44], and glutamatergic [45] transmission. Our present data are consistent with a damaging effect of nicotine on postnatal motor networks even in the spinal cord as long as a critical drug concentration is reached.

### 3.2. Nicotine, Wnt Signaling, and Neurotoxicity

A number of former studies have indicated crosstalk between nicotine and the Wnt signaling pathway. For example, nicotine has been shown to prevent P19 cell differentiation into cardiomyocytes [46], and to induce epithelial–mesenchymal transition [47] and lung cancer [48] via overexpression of the Wnt signaling pathway. In response to stress caused by smoking, suppression of the Wnt/β-catenin pathway has been reported in the airway epithelium [49]. Our present data also support a functional dialogue between Wnt1 and nicotine as low levels of nicotine (0.5 µM) caused a marked increase in Wnt1 activation in parallel with significant neuroprotection. Indeed, the expression of the Wnt1 signal at the ventral spinal cord region was mostly colocalized with S100 signal, a calcium-binding protein mainly expressed by challenged glia [50]. Additionally, upregulation of Wnt1 and β-catenin and attenuation of inducible nitric oxide synthase (iNOS) expression can be associated with neuroprotection in a mouse model of Parkinson’s disease [51]. 

Several research groups have linked Wnt signaling pathways to the inflammatory response that activates astrocytes [27,52], dendritic remodelling after chronic pain models [53] and even neurogenesis in the developing spinal cord [54]. Here, treatment with high dose nicotine caused astroglia activation by the overexpression of GFAP and S100 perhaps indicative of accelerated maturation of astrocytes due to activation of nAChRs [38]. It seems, therefore, likely that astrocytes reacted to the nicotine challenge and contributed to stimulate Wnt1 activity whose reporter gene was relatively well correlated to the glial biomarker S100, typically enhanced after the damage evoked by excitotoxic insult [50].

Although nicotine shows neurotoxic effects on spinal neurons (central and ventral region) and motoneurons, it produced minimal toxicity to dorsal horn neurons, probably due to the high density of neurons [55] and their neuronal nAChRs [56,57,58], which can evoke potent upregulation of inhibitory synaptic activity in the dorsal horn [58] with likely downregulation of overexcitation and consequent damage limitation. 

### 3.3. Effect of High Nicotine on Kainate-Mediated Excitotoxicity

KA was previously used in our laboratory as an excitotoxic agent to induce experimental SCI in a newborn rodent model in vitro that allows monitoring early pathophysiological events for up to 24 h [59,60]. In particular, transient (1 h) KA (50 µM) incubation was shown to abolish fictive locomotor patterns irreversibly and, thereby, provides a model with slowly developing damage and relatively delayed outcome in terms of network structure and function [2,59]. In the present report nicotine (10 µM) co-applied with KA provided significant histological preservation of neurons only in the dorsal horn. At the same time, Nic 10 treatment did not protect central and ventral horn neurons against KA excitotoxicity (Figure 8). Interestingly, even when KA and nicotine were co-applied and the physiological outcome was very poor, a substantial number of neurons was histologically intact. This observation accords with our former results that have indicated a ceiling to excitotoxic damage [59] in line with clinical SCI damage which is rarely total. Nonetheless, despite surviving neurons in large numbers, complex network activities like fictive locomotion collapsed because the network membership of the central pattern generator presumably fell below a critical value whereby coordinated activity becomes impossible [61].

Since KA did not affect the level of the Wnt1-dependent gene expression, the present data indicate that toxicity induced by a glutamate agonist or a cholinergic agonist involved distinct cellular processes, even if they were not additive.

### 3.4. Advantages and Limitations of the Experimental Model and the Study

Since the rodent isolated spinal cord has been used as a classical tool for studying the structure, physiology and pathophysiology of spinal networks [35,36], it may be useful to summarize its experimental advantages and disadvantages. On the plus side, this preparation offers unrivalled access to identified circuits like those for locomotor activity, long survival in vitro (for >24 h) and the opportunity to investigate early development and maturation of spinal networks [55]. On the minus side, it should be noted that this preparation is immature, lacks the role of descending brain inputs, and its in vitro maintenance cannot perfectly reproduce the in vivo condition. For these reasons, data obtained in vitro should be validated with in vivo experiments.

## 4. Materials and Methods

### 4.1. Wild-Type Rat Spinal Cord Preparation

In vitro spinal cord preparations were obtained from postnatal (P0–P2) wild-type Wistar rats (total *n* = 115, where n is the total number of rats used in the study) after decapitation under urethane anesthesia (0.2 mL i.p. of a 10% *w*/*v* solution). Whole spinal cord preparations were gently removed in oxygenated Krebs solution (in mM; 113 NaCl, 4.5 KCl, 1 MgCl_2_.7H_2_O, 2 CaCl_2_, 1 NaH_2_PO_4_, 25 NaHCO_3_, 11 glucose; gassed with 95% oxygen (O_2_) and 5% carbon dioxide (CO_2_); pH 7.4 at room temperature, flowing at 7.5 mL/min) as reported earlier [2,62,63] and were kept in Krebs solution for 2 h to reach functional recovery.

### 4.2. Transgenic Mice Spinal Cord Preparation

Wnt1Cre2 mice (Jackson stock Nº 022501) were mated with Rosa26-tdTomato mice (Rosa26Tom; Jackson stock no. 007909). In mouse genetic lineage tracing strategies, two strains of transgenic mice are mated. One of them carries a transgene that allows the expression of the Cre under a specific promoter (in this case, Wnt1). The other strain contains two loxP sites flanking a STOP signal, before a reporter gene sequence (in this case, Tomato), under a ubiquitously active promoter (Rosa26). When both transgenes recombine in F1, Cre cleaves the loxP sites and removes the STOP signal, so that only in cells in which Wnt1 was activated at some point in development is the reporter gene expressed. Genotyping of Wnt1-Cre2 transgenic mice was performed by PCR with primers (forward, 5′-CAGCGCCGCAACTATAAGAG3′) and (reverse, 5′-CATCGACCGGTAATGCAG 3′) giving a 400 bp product [64]. Wnt1Cre2:Rosa26Tom mice were mated with other Wnt1Cre2:Rosa26Tom animals. The penetration and recombination of both transgenes was assessed by direct observation of tail tip under fluorescent microscope (Appendix A). Only tissues that showed Tomato expression were included and were randomly distributed among the experimental groups. Thoracolumbar spinal cord preparations were isolated from postnatal Wnt1Cre2:Rosa26Tom mice (1–3 days old) in accordance with standard procedures [59]. Protocols to maintain mouse spinal preparations were similar to those for the rat one. Thereafter, spinal cord tissue was immediately fixed in 1× phosphate-buffered saline (1× PBS) containing 4% paraformaldehyde (PFA; 24 h at 4 °C) followed by 30% sucrose 1× PBS for cryoprotection (24 h at 4 °C). After immunostaining, images were taken at a 20× magnification using a Nikon Eclypse NiE microscope (Melville, NY, USA). Since the staining was diffuse, data quantification of Tom expression was performed in terms of immunofluorescence intensity (expressed in arbitrary units, AU) by the ImageJ software (National Institutes of Health (NIH), Bethesda, MD, USA, https://imagej.nih.gov/ij/index.html, accessed on 1 March 2021) in a 150 × 150 µm^2^ area. To measure the colocalization between Wnt1Cre2:Rosa26Tom, NeuN and S100 signals were analyzed at the ventral spinal cord region by Pearson’s correlation were calculated using Colocalization Threshold plugin (from ImageJ software, NIH, accessed on 1 March 2021, https://imagej.nih.gov/ij/index.html). The method was implemented in Costes et al. (2004) [32] using the methodology as reported by Dunn et al. (2011) [31] (Appendix A).

### 4.3. Protocol for Drug Application and Lesioning the Spinal Cord

Nicotine (4 h bath application) was used at 0.5–10 µM concentration (prepared in Krebs solution), the latter previously found to protect brainstem hypoglossal motoneurons from excitotoxic death [1]. It was applied alone or in combination with excitotoxic conditions: data were compared with results from untreated spinal cords were maintained in vitro for 24 h and designated as a sham. 

Kainate (KA, 50 µM, 1 h bath application) was used to induce an excitotoxic lesion primarily affecting the gray matter [4,59], which fully blocks fictive locomotor patterns (evoked either chemically or electrically) for at least 24 h. To investigate the effect of nicotine neuroprotection on spinal locomotor networks, nicotine (10 µM) was applied together with KA (for 1 h) followed by nicotine alone for 4 h (represented as KA+Nic/Nic 4 h, bath application).

To compare the dose-dependent effect of nicotine on network depolarization (measured from the baseline DC level) various nicotine concentrations were used such as 0.5, 1, 2, 10 µM on day 1 (for 4 h, bath application, Figure 9). Changes in DC polarization of any VR represented the summated output of motoneuron population depolarization produced by direct action on such cells plus indirect excitation from pre-motoneurons. After leaving preparations overnight in Krebs solution, reflex activities (mono- and poly-synaptic) and fictive locomotion were tested on the next day as previously reported [4,59].

For Wnt1Cre2:Rosa26Tom mouse experiments the total number (n) of mice used was 42. Four experimental groups were used: sham, Nic 4 h, KA, and KA+Nic/Nic 4 h. Spinal cords were dissected out from mice and kept in Krebs solution for approximately 30 h before fixation. All the procedures were performed with continuous supply of 95% O_2_ and 5% CO_2_ before fixation. The 30 h timepoint was chosen to match the mouse spinal cord preparation with the rat preparation used to perform electrophysiology before fixation. 

From animals showing high efficiency of transgene recombination, different spinal cords preparations were subjected to different experimental conditions. Some spinal cords were treated only with irinotecan (Wnt signaling and DNA topoisomerase I inhibitor, 5 µM, [33,34] or different doses of nicotine (0.5, 1, and 10 µM) and subjected to the same experimental procedures and were processed for immunostaining (Figure 10).

### 4.4. Electrophysiology

Full details of electrophysiological recording were previously reported by Marchetti et al. (2001) [39] and Taccola et al. [4]. Briefly, to investigate the reflex activity and fictive locomotion from the rat spinal cord, VRs of the lumbar (L2 and L5) segments were sucked with tight-fitting suction electrodes to record DC-coupled responses from L2 and L5 VRs which mainly carry flexor and extensor motor responses to the hindlimb muscles, respectively [36]. VR signals were evoked by square pulses (0.1 ms) applied to ipsi-lateral dorsal roots (DR) using bipolar suction electrodes. DR stimulus intensity was adjusted to induce monosynaptic reflex responses, which was considered equivalent to 1× threshold [39] and polysynaptic reflex responses when the stimulus was three times higher [65] as shown in the scheme (Figure 11). The responses were computed by considering 3–5 averaged events for the peak amplitude and area. Previously reported intracellular recordings from spinal motor neurons have confirmed the functional identification of these responses as monosynaptic and polysynaptic reflexes, respectively [66]. D-DRPs were elicited by electrical stimulation of a single L DR and the output signal was recorded from the ipsilateral adjacent DR. To evoke electrically or chemically induced fictive locomotion, a train of 30 pulses at 2 Hz, or bath application of N-methyl-D-aspartate (NMDA; 3–6 µM) plus 5-hydroxytryptamine (5-HT; 10 µM) were used, respectively [36]. Fictive locomotion was characterized by VR rhythmic cycles alternating between left and right side at segmental level and between flexor (L2) and extensor (L5) VRs on the same side once about every 2–3 s [36]. For the analysis of cycle peak amplitude and periodicity, 20 consecutive oscillations were considered as reported previously [4]. Data were acquired, digitized and recorded in pClamp 9.2 (Molecular Devices, Sunnyvale, CA, USA) at 20 KHz (acquisition frequency).

### 4.5. Immunohistochemistry

After completing the electrophysiological experiments, spinal cords were fixed in 4% PFA overnight and then cryoprotected with 30% (*w*/*v*) sucrose the subsequent day. The whole procedure was performed as reported previously [55,63,67,68,69,70]. Transverse spinal sections (30 μm) were cut using a microtome (at –20 °C) from T13 to L5 segments and collected in 1× phosphate buffer solution (PBS) until further use. Spinal cords were processed using a free-floating immunofluorescence protocol where the sections were again washed using 1× PBS followed by incubation with the blocking solution (5% fetal bovine serum, FBS or normal goat serum, NGS; 5% bovine serum albumin, BSA; 0.3% Triton X-100; 1% PBS) at room temperature. Later, spinal sections were immunolabelled with primary antibodies such as anti-SMI32 (specific to non-phosphorylated neurofilament-H of spinal cord motoneurons; mouse monoclonal; 1:1000 dilution; Chemicon, Millipore, Milan, Italy; NE1023), anti-NeuN (specific neuronal marker; rabbit polyclonal; 1:250 and 1:500 dilution; Merck Millipore, Milan, Italy; ABN78), anti-GFAP (specific for glial fibrillary acidic protein, GFAP, member of class III protein family of intermediate filament, expressed in astrocytes and some other astroglial cells in the central nervous system; mouse monoclonal; 1:500; Sigma-Aldrich, Milan, Italy; G3893), anti-S100 (specific for S100 protein which is a part of Ca2+ binding proteins family in astrocytes’ nuclei and cytoplasm, rabbit polyclonal; 1:1000 and 1:400 dilution; DAKO, Glostrup, Denmark; Z0311) and anti-CD31 (specific for Platelet Endothelial Cell Adhesion Molecule-1, mouse monoclonal, Invitrogen, BD Biosciences, Buenos Aires, Argentina; 1:200) at 4 °C for overnight. Antibody validation was performed as reported in our previous studies [2,50]. On the following day, sections were again washed and immunolabeled with secondary antibodies (1:500 dilution; Invitrogen, Carlsbad, CA, USA) such as goat anti-mouse Alexa Fluor 488 (A11029) or 594 (A11032) and goat anti-rabbit Alexa Fluor 488 (A11034) or 594 (A11037) and DAPI (biomarker for cell nuclei; 1:200 dilution used for rat and 1:1000 used for mice; Sigma-Aldrich) for 2 h at room temperature. Samples were directly viewed with a confocal microscope Leica DM6000, FV300 (Olympus Optical, Tokyo, Japan) or Nis-Eclipse microscope (NIKON, Amsterdam, Netherlands) with 20× magnification. Images were captured from the dorsal, central (D, C; 350 × 350 μm^2^) and ventral (V; 300 × 230 μm^2^) spinal regions (in the case of Wistar rats) [2,59,68]. Data were quantified using ImageJ or Fiji software (https://imagej.net/software/fiji/, accessed on 1 March 2021). In the case of Wnt1-Cre2:Rosa26Tom mice, NeuN positive cells were counted in the ventral region. DAPI nuclear staining was used to identify the dead/dying cells quantified with Fiji software. The average percent values of nuclei with condensed chromatin were compared between all experimental groups for the ventral region and normalized to the total number of nuclei. The occurrence of cells with pyknotic nucleus for control condition or after KA application were similar results as previously described for the ventral region [71].

### 4.6. Drugs Used

Drugs used in the experiments include nicotine from Sigma Aldrich (Saint Louis, MO, USA), kainate from Sigma Aldrich (Saint Louis, MO, USA), N-methyl-D-aspartate (NMDA) from Tocris Bioscience (Bristol, UK), 5-hydroxytryptamine (5HT) from Sigma Aldrich (Saint Louis, MO, USA) and irinotecan from Sibudan (Buenos Aires, Argentina).

### 4.7. Statistics

All the data were statistically analyzed using SigmaPlot 14 and SigmaStat (SigmaStat 3.1, Systat Software, Chicago, IL, USA). Data values were indicated as mean ± SEM where mean was represented by thick red bars, *n* represents the number of spinal cords used and N shows the number of spinal slices/spinal cord. Normality test was used to differentiate between parametric and non-parametric data. Parametric values were evaluated either by Welch’s *t-*test (assuming unequal population variances and/or sample sizes) and non-parametric by Mann–Whitney test. One-way analysis of variance test (ANOVA) was used for multiple comparisons and non-parametric data Kruskal–Wallis (either Holm–Sidak or Dunn) test was applied. The accepted significance level was *p* ≤ 0.05 (*** *p* ≤ 0.001, ** *p* ≤ 0.01, * *p* ≤ 0.05).

## 5. Conclusions

Various efforts have been made to induce neuroprotection of spinal neurons against spinal injury [68,72,73,74]. In addition, substantial data are available on nAChRs as a potential target to rescue from neurodegeneration using nicotine [1,2,3]. However, until now little was known what effect a high concentration of nicotine might have on spinal neuronal networks. The present study shows that exposure to a high level of nicotine could cause neuronal death and upregulate the expression of proteins associated with astrocyte activation. Likewise, our results suggest that changes in the levels of activity of Wnt1 modulated by nicotine were accompanied with either neuroprotection or neurotoxicity. Finally, this study highlights the differential vulnerability of the central nervous system of newborn mammals to nicotine exposure, which could irreversibly damage a significant number of spinal neurons. Future work is needed to investigate the long-term effects of high doses of nicotine on the postnatal/fetal spinal locomotor circuitry in vivo.

## Figures and Tables

**Figure 1 ijms-22-09572-f001:**
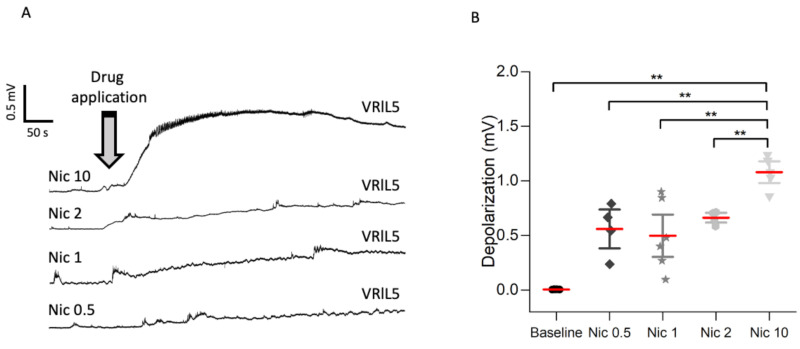
VR depolarization responses induced by applying different nicotine (Nic) doses to the rat spinal cord. (**A**) Sample recordings with Nic 10 (*n* = 5), 2 (*n* = 4), 1 (*n* = 6) and 0.5 (*n* = 4); μM where a grey arrow designates the beginning of drug administration. (**B**) Whisker plot (where mean bars are represented in red) shows baseline (*n* = 5) and depolarization responses induced by Nic 0.5, 1, and 2 with large effect evoked by Nic 10 in comparison to lower Nic doses (Nic 0.5 vs. Nic 10: ** *p* = 0.01, *t*_4.818_ = −3.813, Welch’s *t-*test; Nic 1 vs. Nic 10: ** *p* = 0.005, *t*_7.337_ = −3.987, Welch’s *t-*test; Nic 2 vs. Nic 10: ** *p* = 0.002, *t*_5.486_ = −5.706, Welch’s *t-*test). Note that *n* is the number of spinal cords used.

**Figure 2 ijms-22-09572-f002:**
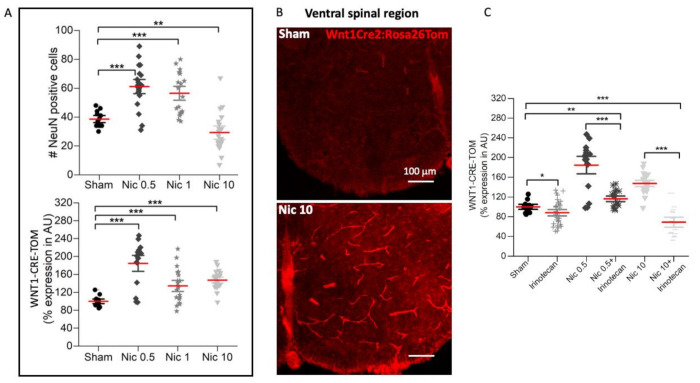
Effect of different doses of nicotine on neuronal numbers and Wnt1 pathway in mouse spinal ventral region. (**A**) Top panel: no change in neuronal numbers after low nicotine doses (0.5 and 1 μM) while a significant decrease was observed with 10 μM nicotine in comparison to sham (Nic 10, A upper panel; sham vs. Nic 10: ** *p* = 0.01, U = 53.5, Mann–Whitney test). Lower panel shows effects of nicotine on Wnt1 immunolabeling whereby Nic 0.5 intensified Wnt1 expression (sham vs. Nic 0.5: *** *p* ≤ 0.001, *t*_17.487_ = −6.83, Welch’s *t-*test) while Nic 1 (Nic 0.5 vs. Nic 1: ** *p* = 0.002, *t*_28.05_ = 3.452, Welch’s *t-*test) or 10 elicited a clearly smaller response (Nic 0.5 vs. Nic 10: ** *p* = 0.009, *t*_18.95_ = 2.94, Welch’s *t-*test). (**B**) Example images of Wnt1 signal in the ventral region for control condition and after nicotine application where Nic 10 application shows an increase in Wnt1 expression. (**C**) Irinotecan significantly decreased Wnt1 expression (sham vs. Irinotecan: * *p* = 0.038, *t*_38.62_ = 2.148, Welch’s *t-*test), prevented the rise by Nic 0.5 (Nic 0.5 vs. Irinotecan+Nic 0.5: ** *p* = 0.003, *t*_19.19_ = 3.42, Welch’s *t-*test), and lowered it below control after Nic 10 (sham vs. Irinotecan+Nic 10: *** *p* = 0.001, *t*_21.87_ = 4.11, Welch’s *t-*test; Nic 10 vs. Irinotecan+Nic 10: *** *p* ≤ 0.001, *t*_26.495_ = 9.82, Welch’s *t-*test). Sham: *n* = 3; Nic 0.5: *n* = 5; Nic 1: *n* = 5; Nic 10: *n* = 5; Irinotecan, *n* = 8; Nic 10+Irinotecan, *n* = 4, where n is the number of spinal cords used. Note that mean bars in the whisker plots are represented in red. 3–4 slices were analyzed per spinal cord. Scale bar (100 μm) applies to all panels.

**Figure 3 ijms-22-09572-f003:**
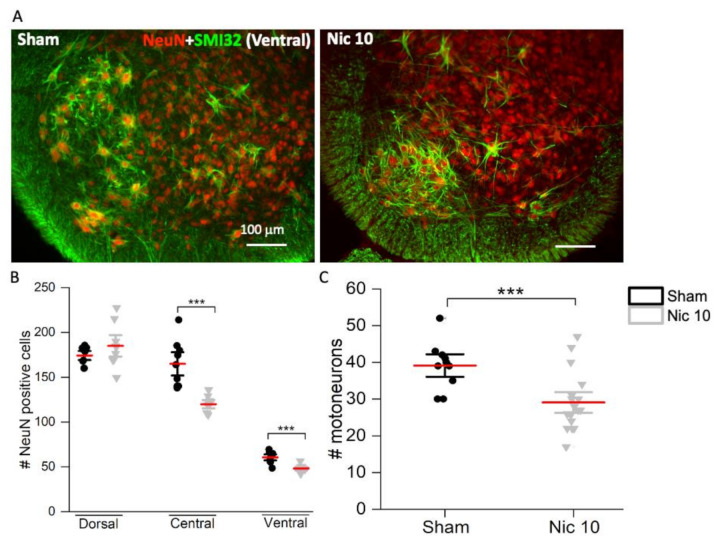
Changes in neuronal and motoneuronal numbers after incubation with Nic 10 (*n* = 3) in rat spinal cord. (**A**,**B**) No modulation in the number of neurons (immunolabelled by NeuN, red in panel **A**) in the dorsal horn (D; sham vs. Nic 10: *p* = 0.238, *t*_10.781_ = −1.249, Welch’s *t-*test) whereas significant reduction in the central (C; sham vs. Nic 10: *** *p* ≤ 0.001, *t*_10.034_ = 4.903, Welch’s *t-*test) and the ventral (V; sham vs. Nic 10: *** *p* ≤ 0.001, *t*_13.722_ = 4.658, Welch’s *t-*test) regions occurred (*n* = 3). (**A**,**C**) Spinal motoneurons immunolabelled by SMI32 were substantially decreased after Nic 10 application (*** *p* = 0.001, *t*_22.002_ = 3.637, Welch’s *t-*test). Note that mean bars in the whisker plots are represented in red. A minimum of three slices was analyzed per spinal cord, therefore nine slices were analyzed per treatment. Scale bar (100 μm) applies to all panels.

**Figure 4 ijms-22-09572-f004:**
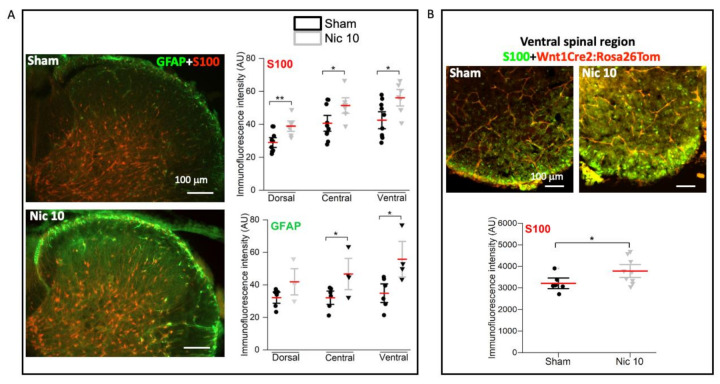
Glial effects by 10 μM Nicotine. (**A**) in the D, C and V regions of the rat spinal cord of rats S100 expression was raised (sham: *n* = 10 and Nic 10: *n* = 7; D: ** *p* = 0.004, *t*_14.08_ = −3.464; C: * *p* = 0.03, *t*_14.464_ = −2.404; V: * *p* = 0.013, *t*_14.486_ = −2.843; Welch’s *t-*test) and similar rise was observed with GFAP, (sham: *n* = 6 and Nic 10: *n* = 4; D: *p* = 0.09, *t*_8_ = −1.907; C: * *p* = 0.04, *t*_8_ = −2.385; V: * *p* = 0.02, *t*_8_ = −2.795; Welch’s *t-*test) biomarkers. (**B**) Similar over-expression of S100 was observed in the ventral spinal cord of mice (* *p* = 0.044, *t*_12.983_ = −2.224, Welch’s *t-*test). Note that mean bars in the whisker plots are represented in red. Scale bar (100 μm) applies to all panels.

**Figure 5 ijms-22-09572-f005:**
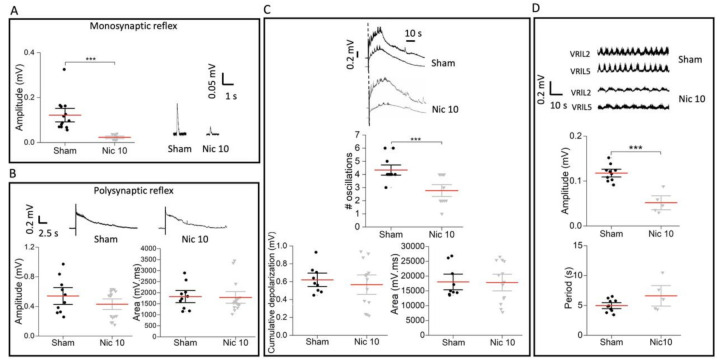
Limited/impaired synaptic transmission and fictive locomotion after Nic 10 (*n* = 5) in the rat spinal cord. (**A**,**B**) Nic 10 abolished monosynaptic reflexes (sham vs. Nic 10: *** *p* ≤ 0.001, U = 0, Mann–Whitney test) whereas polysynaptic reflexes were not significantly affected in comparison to sham (*n* = 10) (sham vs. Nic 10: Amplitude: *p* = 0.445, U = 65; Area: *p* = 0.617, U = 70, Mann–Whitney test). (**C**,**D**) DR evoked oscillations (number of oscillations: *** *p* ≤ 0.001, *t*_22.66_ = 3.943, Welch’s *t-*test), their amplitude and period were significantly decreased by Nic 10 (amplitude: *** *p* = 0.001, *t*_6.588_ = 5.548; period: *p* = 0.09, *t*_13_ = −1.801; Welch’s *t-*test) with no change in cumulative depolarization amplitude or area. Note that mean bars in the whisker plots are represented in red.

**Figure 6 ijms-22-09572-f006:**
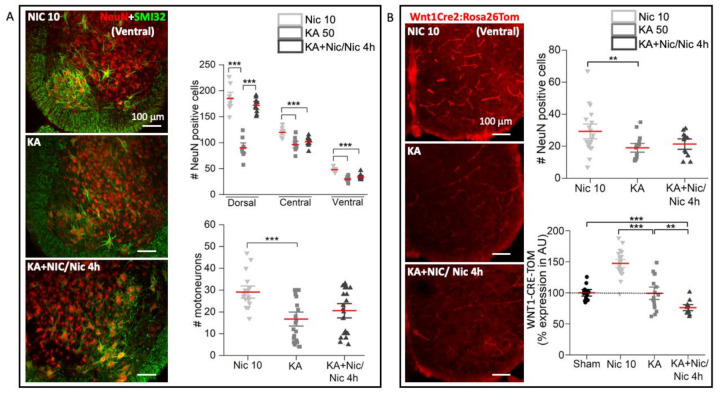
Neuronal number and Wnt1activity after KA application alone or with the co-application protocol KA+Nic 10 followed by Nic 10 for 4 h (KA+Nic/Nic 4 h). (**A**) KA administration caused fall in neuronal (NeuN red in the example images; Nic: *n* = 3, KA: *n* = 3, KA+Nic/Nic 4 h: *n* = 3) (Nic 10 vs. KA: *** *p* ≤ 0.001, *t*_15.228_ = 9.337, Welch’s *t-*test) and motoneuronal (SMI32 green; Nic 10: *n* = 6, KA: *n* = 7, KA+Nic/Nic 4 h: *n* = 7) (Nic 10 vs. KA: *** *p* ≤ 0.001, *t*_37.95_ = 4.361, Welch’s *t-*test) numbers while Nic 10 application together with KA preserved neurons in the dorsal horn (KA vs. KA+Nic/Nic 4 h: *** *p* ≤ 0.001, *t*_15.03_ = −10.21, Welch’s *t-*test) with no improvement in the C (KA vs. KA+Nic/Nic 4 h: *p* = 0.35, *t*_14.39_ = −0.974, Welch’s *t-*test) and V (KA vs. KA+Nic/Nic 4 h: *p* = 0.1, *t*_15.8_ = −1.73, Welch’s *t-*test) region in the rat spinal cord. No recovery in motoneuronal numbers were observed after KA+Nic/Nic 4 h (KA vs. KA+Nic/Nic 4 h: *p* = 0.2, *t*_41.96_ = −1.248, Welch’s *t-*test) in the rat spinal cord. (**B**) In the mouse spinal cord, KA (KA: *n* = 4; Nic 10: *n* = 5) alone significantly diminished neurons in the ventral horn (KA vs. Nic 10: ** *p* = 0.007, *t*_30.05_ = 2.887, Welch’s *t-*test) with no recovery observed after KA+Nic/Nic 4 h (*n* = 3) (KA vs. KA+Nic/Nic 4 h: *p* = 0.42, *t*_26_ = −0.83, Welch’s *t-*test) treatment. KA application did not change the % Wnt1 expression level in comparison to sham (sham vs. KA: *p* = 0.910, *t*_22.052_ = 0.114, Welch’s *t-*test) which is shown by a black dotted line (as 100%) in the bar plot (lower panel). However, Nic 10 application together with KA significantly reduced the % Wnt expression (KA vs. KA+Nic/Nic 4 h: ** *p* = 0.005, *t*_21.2_ = 3.13, Welch’s *t-*test). Note that mean bars in the whisker plots are represented in red. Minimum 3–4 slices (N) were analyzed per spinal cord. Scale bar (100 μm) applies to all panels.

**Figure 7 ijms-22-09572-f007:**
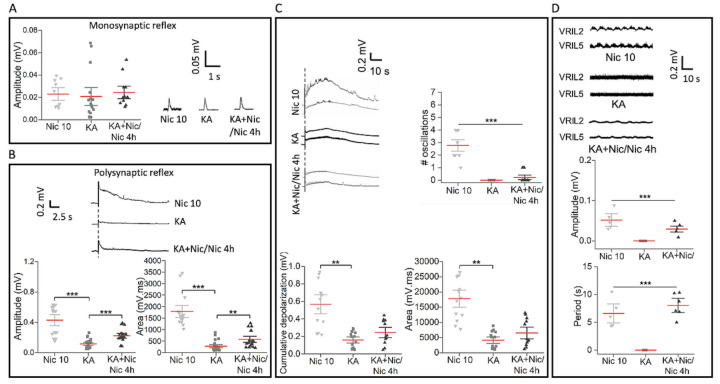
Synaptic transmission and fictive locomotion after excitotoxic insult by KA (*n* = 8) or treatment with co-application of KA+Nic 10 followed by Nic 10 for 4h (KA+Nic/Nic 4 h, *n* = 6) in the rat spinal cord. (**A**) Monosynaptic reflex responses were suppressed by Nic 10 (*n* = 5) which were similar to the responses evoked by KA with no further improvement after KA+Nic/Nic 4 h application (*p* = 0.2, H_(2)_ = 3.108, Kruskal–Wallis one-way analysis of variance on ranks test). (**B**) Polysynaptic reflex responses diminished by KA (Nic 10 vs. KA; amplitude: *** *p* ≤ 0.001, *t*_17.052_ = 6.34, Welch’s *t-*test; area: *** *p* ≤ 0.001, *t*_17.1_ = 8.350; Welch’s *t-*test) were partly recovered by Nic 10 when co-applied with KA (KA vs. KA+Nic/Nic 4 h; amplitude: *** *p* ≤ 0.001, *t*_28.34_ = −4.372; area: ** *p* = 0.006, *t*_24.58_ = −3.013; Welch’s *t-*test). (**C**) Electrically induced fictive locomotion blocked by KA (Nic 10 vs. KA; number of oscillations: *** *p* ≤ 0.001, *t*_12_ = 9.144; cumulative depolarization: *** *p* ≤ 0.001, *t*_14.51_ = 5.38; area: ****p* ≤ 0.001, *t*_15.41_ = 6.83; Welch’s *t-*test) was not recovered after Nic 10 co-administration. (**D**) Chemically induced fictive locomotion disrupted by KA (amplitude: ** *p* = 0.002, U = 0; period: ** *p* = 0.002, U = 0; Mann–Whitney test) was very mildly recovered after Nic 10 co-application with KA followed by Nic 4 h (KA vs. KA+Nic/Nic 4 h; amplitude: *** *p* ≤ 0.001, U = 0; period: *** *p* ≤ 0.001, U = 0; Mann–Whitney test). Note that mean bars in the whisker plots are represented in red.

**Figure 8 ijms-22-09572-f008:**
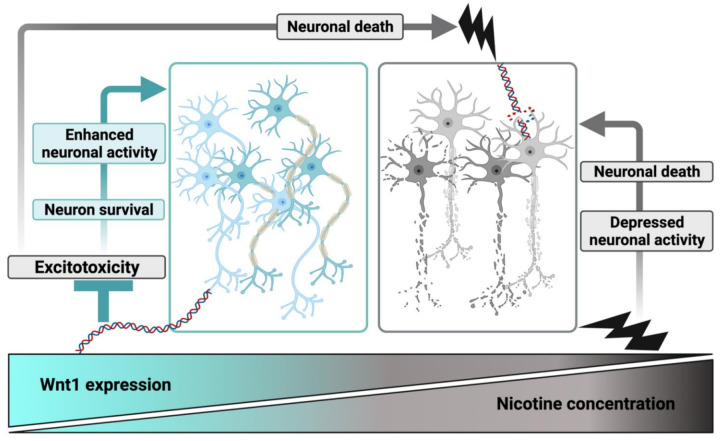
Summary of interactions among nicotine, Wnt1 pathway and excitotoxicity in spinal networks of postnatal rodents. High nicotine concentration led to low Wnt1 expression, which occurred together with depressed neuronal activity and finally neuronal death. Conversely, low nicotine dose led to increased Wnt1 expression and also rescued neurons from excitotoxic cell death (by contrasting excitotoxic mechanisms) with the outcome of neuronal survival and higher neuronal activity.

**Figure 9 ijms-22-09572-f009:**
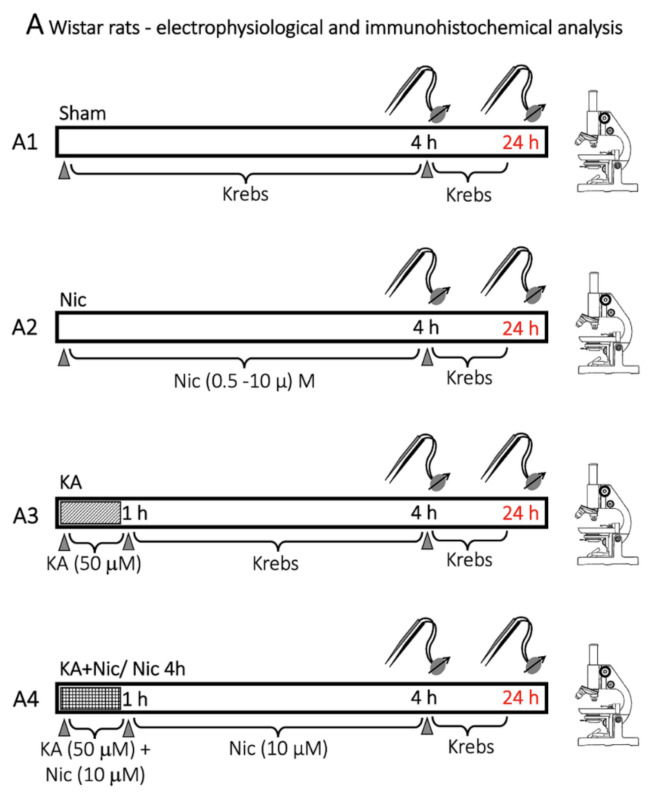
Protocol for drug application, electrophysiological recordings (Elephys) and immunohistochemistry performed with wild type Wistar rat spinal cord (P0–P2).

**Figure 10 ijms-22-09572-f010:**
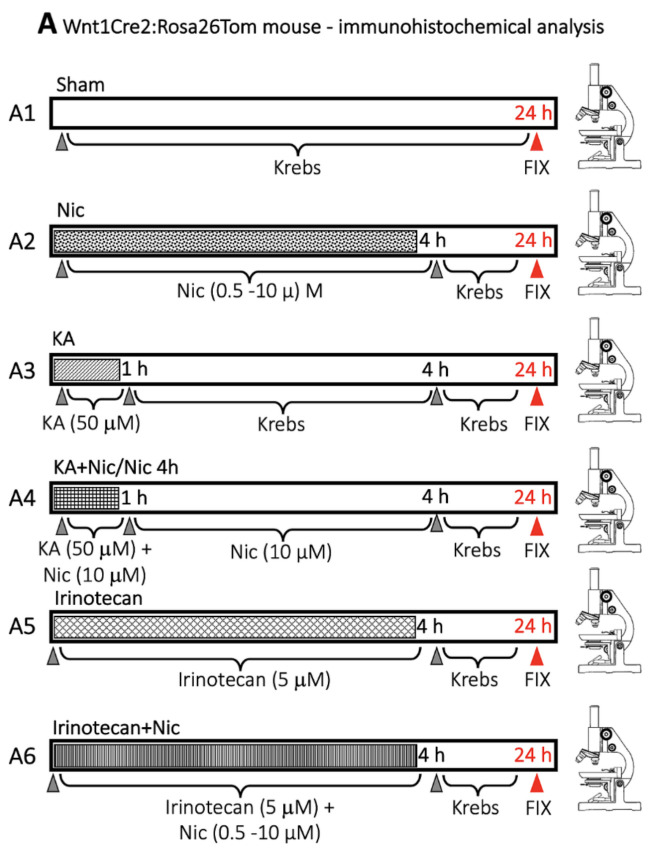
Protocol for drug administration and immunohistochemistry performed with Wnt1Cre2:Rosa26Tom mouse spinal cord.

**Figure 11 ijms-22-09572-f011:**
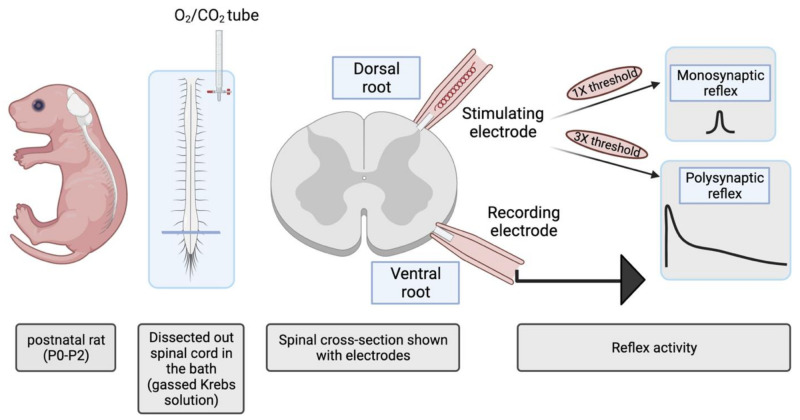
Schematic representation of isolated spinal cord preparation used for recording the reflex activity from the lumbar (L2/L5) ventral roots of the postnatal rat. Monosynaptic and polysynaptic reflexes evoked by electrical stimulation of one dorsal root were recorded from ipsi-lateral and ipsi-segmental ventral roots.

## Data Availability

Data displayed in this study are available on request from the corresponding author.

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
