# Peer review of "Nicotine Neurotoxicity Involves Low Wnt1 Signaling in Spinal Locomotor Networks of the Postnatal Rodent Spinal Cord"

_ijms, 2021, doi:10.3390/ijms22179572_

Round 1

Reviewer 1 Report

The present manuscript was well written and interesting.

Self-citation in References might be rather much.

In line 495-496, "Marchetti et al. (2001) [31]" was an error. The reference [31] was not by Marchetti et al., and Marchetti et al. (2001) was the reference [39].

Author Response

Dear Editor,

We would like to resubmit our revised manuscript “Nicotine neurotoxicity involves low Wnt1 signaling in spinal locomotor networks of the postnatal rodent spinal cord” to be considered for publication as a research article in the International Journal of Molecular Sciences (IJMS) under the special issue “Molecular Research in Neurotoxicology 2.0”. We have systematically revised our manuscript according to the suggestions of the reviewers whose comments have been greatly appreciated. All changes are highlighted as track changes in the text.

Reviewer 1

The present manuscript was well written and interesting.

  1. R: Self-citation in References might be rather much.

AU: We realize that we quoted our previous work many times and if this may appear a kind of self-indulgence, we apologize. However, this was due to the fact we were the first to introduce the isolated spinal cord as a model preparation for studying early changes in spinal networks after experimental injury and, inevitably, we found ourselves to quote our papers in support of the present techniques and basic data.

  1. R: In line 495-496, "Marchetti et al. (2001) [31]" was an error. The reference [31] was not by Marchetti et al., and Marchetti et al. (2001) was the reference [39].

AU: Corrected.

Reviewer 2 Report

Overall, this was a very nice study.  I have a few minor comments.  For all statistical test computed, the effects sizes are not listed and should be reported and explained within the context of each experiment.  

Further, some method sections are lacking detail and sentences should not begin with acronyms.  Please elaborate in more detail what was done (e.g., for the drugs, how and why were they used, and at what concentrations for how long, etc.).

Lastly, a model figure illustrating how the electrophysiology was conducted for both the monosynaptic and polysynaptic reflex preparations would be helpful for the reader to follow.

Author Response

Thank you. We have systematically revised (changes are highlighted as track changes) our manuscript according to the suggestions of the reviewers whose comments have been greatly appreciated. All changes are highlighted as track changes in the text.

Reviewer 2

Overall, this was a very nice study.  I have a few minor comments. 

  1. R: For all statistical test computed, the effects sizes are not listed and should be reported and explained within the context of each experiment.  

AU: Reported as a percent change or fold (increase/ decrease) in the result section.

  1. R: Further, some method sections are lacking detail and sentences should not begin with acronyms. 

AU: Corrected.

  1. R: Please elaborate in more detail what was done (e.g., for the drugs, how and why were they used, and at what concentrations for how long, etc.).

AU: More details of drugs application including the concentration, how, why and how long they were used is included in materials and methods section 4.3 “Protocol for drug application and lesioning the spinal cord”. Further clarification is provided by three new Figs (9, 10, 11) that indicate the time course of our procedures (as also requested by another reviewer).

  1. R: Lastly, a model figure illustrating how the electrophysiology was conducted for both the monosynaptic and polysynaptic reflex preparations would be helpful for the reader to follow.

AU: Added in the materials and methods section as Figure 11.

Reviewer 3 Report

The manuscript IJMS-1338937 “Nicotine neurotoxicity involves low Wnt1 signaling in spinal locomotor networks of the postnatal rodent spinal cord” attempts to describe the effects of nicotine in three different concentrations, and the effects of the excitotoxin on neurons.  The authors state that low dose nicotine (1μM) is neuroprotective, but higher doses can be neurotoxic.  They use Wistar rats as their model system and the Wnt1Cre2:Rosa26Tom double-transgenic mouse. The authors conclude that that high dose nicotine is neurotoxic to central and ventral spinal neurons as the neuroprotective role of Wnt signaling became attenuated. This also corroborates the risk of nicotine exposure in the fetus/newborn.

Well written and well designed.  There are a few suggestions, mostly for clarity.

I would recommend – based on the size of the figures that the whisker plot be used with the median or mean bar – and not the bar graph.  Or simply use the bar graph and state what the variance was in the figure legend.  The bars in their current state are too narrow to add in the individual data symbols.  In the current state, they seem a bit cluttered.

The photomicrographs are very nice and add a lot to the submission, but they too are very small – if there is anyway to change the size of the photomicrograph, maybe 40-50% bigger, it will make it easier to see the location and  colocalization of the neurons.

The statistics section is very well explained.  Clear and appropriate for the data being presented. A couple of things – t-test should be hyphenated.  This occurs multiple times throughout so a global search and replace may be in order.

If possible, a schematic diagram would add tremendously to the information and the clarity of the presentation.  Similar to a graphical abstract, where the reader can graphically see what the authors are describing since there is a lot happening here.  We have locomotor activity, electrophysiology, histochemistry, etc.  This may help the reader understand the pathways and mechanisms involved.

MINOR (with LINE NUMBER)

24 – ‘suggesting their response to stress’ is a bit awkward.  Consider “indicating a stress response”

29 – what does ‘this also’ refers too?  The toxicity of high dose nicotine, or the suppression of Wnt1 or both?

41 - change ‘human’ to humans

50 – is it necessary to use ‘proconvulsive’?  Wouldn’t it just be ‘convulsive’?

62 – change ‘depend’ to depends

70 – delete ‘an’ before ‘early’

78 – insert ‘it’ before appears

91 – the fragment of the sentence “with the purpose of designing” could be changed to be less wordy – consider ‘to design’

104 – hyphenate high-frequency

118 – insert ‘a’ before lesser

163 – delete ‘to’ before ‘prevent’.

174 – insert a comma after ‘motoneuron’ and before ‘particularly’

186 – insert ‘the’ before fetal.  Also check for consistency in the spelling of fetal, or foetal – either way is fine, but needs to be consistent.

287 – insert ‘the’ before application

289 – insert ‘a’ before decrease

290 – insert ‘a’ before similar

302 – delete ‘the’ before one (or possibly just ‘similar to untreated controls’)

338 – hyphenate nicotine-induced

347 – delete “In fact,” to start the sentence – start with “Our genetic model…”

350 – insert ‘an’ before increased

357 – insert ‘the’ before nicotine-evoked

358 – insert ‘the’ before release

368 – insert ‘the’ before presynaptic

376 – delete ‘actually’

378 – delete ‘clearly’

388 – delete ‘a’ before suppression

390 – insert ‘a’ before marked

393 – insert ‘is’ before mainly

404 – insert ‘the’ before damage and the word ‘origin’ doesn’t appear to belong at the end of the sentence.  Should that be ‘organs’?  Regardless this sentence on lines 403 and 404 is a bit confusing.

416 and 419 – coapplied is hyphenated and not hyphenated.  Please standardize.

422 – insert ‘is’ before in

474 – insert ‘were’ before maintained; and then insert ‘a’ before sham

480 – change ‘In order to compare…’ to ‘To compare…” and then hyphenate dose-dependent

488 – delete ‘as follows’ (a little wordy and not needed in this context)

498 – hyphenate DC-coupled

553 – insert ‘the’ before ventral

565 – insert ‘the’ before number – in both instances

569 – delete ‘for’ at the end of the line before non-parametric

Author Response

Thank you. We have systematically revised (changes are highlighted as track changes) our manuscript according to the suggestions of the reviewers whose comments have been greatly appreciated. All changes are highlighted as track changes in the text.

Reviewer 3

The manuscript IJMS-1338937 “Nicotine neurotoxicity involves low Wnt1 signaling in spinal locomotor networks of the postnatal rodent spinal cord” attempts to describe the effects of nicotine in three different concentrations, and the effects of the excitotoxin on neurons.  The authors state that low dose nicotine (1μM) is neuroprotective, but higher doses can be neurotoxic.  They use Wistar rats as their model system and the Wnt1Cre2:Rosa26Tom double-transgenic mouse. The authors conclude that that high dose nicotine is neurotoxic to central and ventral spinal neurons as the neuroprotective role of Wnt signaling became attenuated. This also corroborates the risk of nicotine exposure in the fetus/newborn.

R: Well written and well designed.  There are a few suggestions, mostly for clarity.

AU: We are most grateful to this reviewer for indicating all the minor changes that have helped us to improve our MS. They have all been implemented in the resubmitted text.

  1. R: I would recommend – based on the size of the figures that the whisker plot be used with the median or mean bar – and not the bar graph. Or simply use the bar graph and state what the variance was in the figure legend.  The bars in their current state are too narrow to add in the individual data symbols.  In the current state, they seem a bit cluttered.

AU: Corrected. All figures are modified, and whisker plots are used with mean bars (labelled in red for clarity).

  1. R: The photomicrographs are very nice and add a lot to the submission, but they too are very small – if there is anyway to change the size of the photomicrograph, maybe 40-50% bigger, it will make it easier to see the location and colocalization of the neurons.

AU: The co-localization images of supplemental Fig. 1B were increased in size as suggested. Also, all other photomicrographs in other Figures were increased in size (Fig 2, 3, 4 and 6) although not doubled in size. We had to include four new Figs. If all microphotos should be doubled in size, we would need adding at least four Figs to the MS. We fear that our MS would then be over-illustrated.

  1. R: The statistics section is very well explained. Clear and appropriate for the data being presented. A couple of things – t-test should be hyphenated.  This occurs multiple times throughout so a global search and replace may be in order.

AU: Modified as suggested.

  1. R: If possible, a schematic diagram would add tremendously to the information and the clarity of the presentation. Similar to a graphical abstract, where the reader can graphically see what the authors are describing since there is a lot happening here.  We have locomotor activity, electrophysiology, histochemistry, etc.  This may help the reader understand the pathways and mechanisms involved.

AU: Thank you for this suggestion.  Two schematics diagrams to explain the protocols used were introduced in the materials and methods, section 4.3 “Protocol for drug application and lesioning the spinal cord” as Figure 9 and 10.

A graphical abstract (Figure 8) was introduced to the section 3 of discussion.

MINOR (with LINE NUMBER)

R: 24 – ‘suggesting their response to stress’ is a bit awkward.  Consider “indicating a stress response”

AU: Modified as 'indicating a stress response'

R: 29 – what does ‘this also’ refers too?  The toxicity of high dose nicotine, or the suppression of Wnt1 or both?

AU: This also refers to ‘Toxicity of high dose nicotine

R: 41 - change ‘human’ to humans

AU: Corrected

R: 50 – is it necessary to use ‘proconvulsive’?  Wouldn’t it just be ‘convulsive’?

AU: Corrected

R: 62 – change ‘depend’ to depends

AU: Corrected

R: 70 – delete ‘an’ before ‘early’

AU: Corrected

R: 78 – insert ‘it’ before appears

AU: Corrected

R: 91 – the fragment of the sentence “with the purpose of designing” could be changed to be less wordy – consider ‘to design’

AU: Corrected

R: 104 – hyphenate high-frequency

AU: Corrected

R: 118 – insert ‘a’ before lesser

AU: Corrected

R: 163 – delete ‘to’ before ‘prevent’.

AU: Corrected

R: 174 – insert a comma after ‘motoneuron’ and before ‘particularly’

AU: Inserted

R: 186 – insert ‘the’ before fetal.  Also check for consistency in the spelling of fetal, or foetal – either way is fine, but needs to be consistent.

AU: Inserted ‘the’ before fetal. Also checked for consistency in the spelling of fetal in the whole text and the spelling is consistent.

R: 287 – insert ‘the’ before application

AU: Inserted

R: 289 – insert ‘a’ before decrease

AU: Inserted

R: 290 – insert ‘a’ before similar

AU: Inserted

R: 302 – delete ‘the’ before one (or possibly just ‘similar to untreated controls’)

AU: Modified as ‘similar to untreated controls

R: 338 – hyphenate nicotine-induced

AU: Hyphenated

R: 347 – delete “In fact,” to start the sentence – start with “Our genetic model…”

AU: Modified

R: 350 – insert ‘an’ before increased

AU: Inserted

R: 357 – insert ‘the’ before nicotine-evoked

AU: Inserted

R: 358 – insert ‘the’ before release

AU: Inserted

R: 368 – insert ‘the’ before presynaptic

AU: Inserted

R: 376 – delete ‘actually’

AU: Deleted

R: 378 – delete ‘clearly’

AU: Deleted

R: 388 – delete ‘a’ before suppression

AU: Deleted

R: 390 – insert ‘a’ before marked

AU: Inserted

R: 393 – insert ‘is’ before mainly

AU: Inserted

R: 404 – insert ‘the’ before damage and the word ‘origin’ doesn’t appear to belong at the end of the sentence.  Should that be ‘organs’?  Regardless this sentence on lines 403 and 404 is a bit confusing.

AU: Inserted ‘the’ before damage. Deleted the word ‘origin’ and replaced the end of the sentence to ‘the damage evoked by excitotoxic insult’

R: 416 and 419 – coapplied is hyphenated and not hyphenated.  Please standardize.

AU: Standardized to hyphenated

R: 422 – insert ‘is’ before in

AU: Inserted

R: 474 – insert ‘were’ before maintained; and then insert ‘a’ before sham

AU: Inserted ‘were’ before maintained and ‘a’ before sham.

R: 480 – change ‘In order to compare…’ to ‘To compare…” and then hyphenate dose-dependent

AU: Modified as suggested.

R: 488 – delete ‘as follows’ (a little wordy and not needed in this context)

AU: Deleted

R: 498 – hyphenate DC-coupled

AU: Hyphenated

R: 553 – insert ‘the’ before ventral

AU: Inserted

R: 565 – insert ‘the’ before number – in both instances

AU: Inserted as suggested

R: 569 – delete ‘for’ at the end of the line before non-parametric

AU: Deleted

Reviewer 4 Report

The manuscript presents an interesting study that evaluated the effects of high doses of nicotine on spinal neuronal networks in newborns, highlighting the differential vulnerability of the central nervous system of newborn mammals to nicotine exposure.

The manuscript is well written and interesting for the research community. Some issues have to be solved before the acceptance of the manuscript.

  1. In the material and methods section add the manufacturer for the reagents used.
  2. Define all the abbreviations at the first use in the manuscript
  3. Please specify if the 2 experiments involving the use of rats and mice for obtaining the spinal cord preparations have ethical approval.
  4. A flowchart to represent the study design will be helpful for the reader
  5. In the discussion section, the strengths and limitations of the manuscript should be clearly stated

Author Response

Reviewer 4

The manuscript presents an interesting study that evaluated the effects of high doses of nicotine on spinal neuronal networks in newborns, highlighting the differential vulnerability of the central nervous system of newborn mammals to nicotine exposure.

The manuscript is well written and interesting for the research community. Some issues have to be solved before the acceptance of the manuscript.

1. R: In the material and methods section add the manufacturer for the reagents used.

AU: the text was modified as suggested in material and methods section 4.6 “Drugs used”.

2. R: Define all the abbreviations at the first use in the manuscript

AU: Defined as suggested.

3. R: Please specify if the 2 experiments involving the use of rats and mice for obtaining the spinal cord preparations have ethical approval.

AU: The ethical approval for using both rat and mice has been included after the conclusion under the sub-heading “Ethical statement”.

4. R: A flowchart to represent the study design will be helpful for the reader.

AU: Two schematics (Figure 9 and 10) to explain the protocols used were introduced into the materials and methods, section 4.3 “Protocol for drug application and lesioning the spinal cord”.

5. R: In the discussion section, the strengths and limitations of the manuscript should be clearly stated. 

AU: One paragraph is included at the end of discussion under sub-section “3.4 Advantages and limitations of the experimental model and the study”

Round 2

Reviewer 4 Report

The authors addressed all my comments. The manuscript is much improved and ready for acceptance.